# Strength and Endurance Training in Older Women in Relation to *ACTN3 R577X* and *ACE I/D* Polymorphisms

**DOI:** 10.3390/ijerph17041236

**Published:** 2020-02-14

**Authors:** Cristina Romero-Blanco, María Jesús Artiga-González, Alba Gómez-Cabello, Sara Vila-Maldonado, José Antonio Casajús, Ignacio Ara, Susana Aznar

**Affiliations:** 1Physical Activity and Health Promotion Research Group, Universidad de Castilla-La Mancha, 45004 Toledo, Spain; Cristina.Romero@uclm.es; 2Spanish National Cancer Center (CNIO), 28029 Madrid, Spain; mjartiga@cnio.es; 3Growth, Exercise, Nutrition and Development Research Group, Universidad de Zaragoza, 50009 Zaragoza, Spain; agomez@unizar.es (A.G.-C.); joseant@unizar.es (J.A.C.); 4Growth, Exercise, Nutrition and Development Research Group, Universidad de Castilla-La Mancha, 45071 Toledo, Spain; sara.vila@uclm.es (S.V.-M.); ignacio.ara@uclm.es (I.A.); 5CIBERFES Biomedical Research Networking Center on Frailty and Health Aging, 28029 Madrid, Spain

**Keywords:** *ACE*, *ACTN3*, physical fitness, genotype, women, elderly

## Abstract

The purpose of this study is to analyze the effect of two genetic polymorphisms, *ACTN3*
*R577X*, and *ACE I/D*, on physical condition in a sample of active older women after a two-year training period. The sample was composed of 300 healthy women over the age of 60 who underwent a two-year training program. Adapted tests from the Senior Fitness Test were used. The genotyping of the polymorphisms was obtained from the participants’ DNA via buccal swabs. The analysis of the *ACE* polymorphism did not reveal differences between genotypes. The analysis of the *R577X* polymorphism showed a favorable effect for the *ACTN3 XX* genotype in tests for leg strength (*p*: 0.001) after training, compared to the other genotypes, and also in the analysis of the combined effect of the polymorphism (*ACE II + ACTN3 RX/XX*). The intragroup effect revealed an improvement in arm strength for carriers of the X allele after 24 months of training (*p* < 0.05). The endurance values significantly worsened in all study groups. Conclusions: The *R577X* polymorphism of *ACTN3* may have an important role in capacities related to muscle strength, providing a beneficial effect for carriers of the *X* allele.

## 1. Introduction

Life expectancy among the world’s population is steadily increasing, and the percentage of older people in developed countries is on the rise [1]. However, ageing is not without associated problems, such as a loss of independence [2], a greater risk of falls [3], and an increase of chronic illnesses [4], all of which lead to a decline in quality of life [5]. 

In 2002, the World Health Organization (WHO) established a global strategy to support the active ageing of the older population as a measure to support opportunities of biopsychosocial health among this collective [6]. Since then, numerous studies have concentrated on researching measures for improving the quality of life of older people, several of which are based on the development of physical capacities as a measure of healthy ageing [7,8,9]. Furthermore, other authors have attempted to find a biological explanation [10] underlying two genetic variations which have been considered to be particularly striking: the *R577X* polymorphism (rs1815739) of the *α-actinin 3 gene* (*ACTN3*) and the *I/D* polymorphism (rs1799752) of the *angiotensin-converting enzyme gene* (*ACE*).

*ACTN3* is a structural component protein that predominates in the Z zones of the sarcomere. A very common polymorphism of *α-actinin-3* has been identified in humans in position 1747 of exon 16, wherein arginine is converted into a stop codon *(R577X)* due to a cytosine-to-thymine substitution [11]; as a result, this protein is not expressed. *ACTN3* is almost exclusively expressed in type II (fast twitch) muscle fibers [12], and the deficiency of *α-actinin-3* among the general population appears to be associated with a decrease of the mass, muscle strength and fragility that accompanies ageing [13,14]. However, among the older population, the biological effect of this genetic variant is unclear. Although some studies do not show any type of relation between the loss of physical capacity and this genotype in Caucasian older populations [15], others have reported an association, especially among women; however, there is a lack of agreement on which is the favorable allele [16,17].

In addition, the genetic polymorphism that is best characterized in relation to exercise is the presence or absence (insertion/deletion; *I/D*) of a fragment of 287 pairs of bases (pb) in intron 16 of the *angiotensin-converting enzyme gene* (*ACE*) [18]. This polymorphism has been strongly associated to sports. More specifically, the I allele has been related with resistance and *D* alleles have been related with strength/power [19]. This polymorphism has been associated both with physical condition in older people as well as with longevity [20,21]. The studies performed on the older population also present controversial results regarding the role of the polymorphism *I/D* of *ACE*. In several studies performed among older people, a positive correlation has been found between the presence of the *D* allele and speed, strength, agility and/or resistance [22,23,24]. However, other studies have not found any relation with any of the physical condition variables analyzed [15,25,26]. 

Considering the relation between these two polymorphisms in the field of sports [27] and the few studies available that have analyzed the effect of prolonged training among older people, the purpose of this study was to assess the effect of these two polymorphisms in a group of older active women after a two-year training program, as well as to study the associations between these polymorphisms and physical condition among older women.

## 2. Materials and Methods 

### 2.1. Participants

The study sample was selected from the multi-centric EXERNET population (Research Network in Physical Exercise and Health for Special Populations) [28]. The EXERNET multi-center study is a cross-sectional study on physical fitness and body composition evaluation and its relation with healthy lifestyle among non-institutionalized elderly from six regions in Spain. The population was selected by means of a multi-step, simple random sampling, considering first the location that ensures the geographic and cultural diversity of the sample, then three different cities in each region (the capital of the region and two other cities—one with 10,000–40,000 habitants and another of 40,000–10,000 habitants) and, finally, a random assignment of the civic and sports centers. All participants were of the same Caucasian (Spanish) descent from three or more generations.

Prior to the recruitment of study participants, the informed consent document used in this study was approved by the Clinical Research Ethics Committee (18/08; 1/12). After signing the informed consent, a total of 300 women participated in the current study. 

The participating subjects were selected according to the following inclusion criteria: women aged over 60 years old, who were not institutionalized, who had participated in a local physical activity program in their municipality and had signed the informed consent. 

### 2.2. Anthropometric Measures

The height of participants was determined in centimeters using a portable measuring rod (SECA 711). The weight and percentage of total fat (%MG) were measured using a bioelectric impedance device (Tanita BC 418-MA, Tanita Corp., Tokyo, Japan). Subsequently, the body mass index (BMI) was calculated using the formula BMI = kg/m^2^.

### 2.3. Active–Sedentary Behavior 

The validated EEPAQ (Elderly EXERNET Physical Activity Questionnaire) [29] was used to evaluate the number of hours walking per day of participants and the time dedicated to sedentary activities (number of hours sitting per day).

### 2.4. Assessment of Physical Condition

To evaluate physical condition, adapted tests of the Senior Fitness Test battery were used [7]. We evaluated the strength of upper and lower extremities and endurance. 

The Chair Stand Test was used to register lower extremity strength: i.e., the number of times that the participant was able to sit and stand over a 30 s period.

The Arm Curl Test was used to evaluate upper extremity strength: i.e., the number of flexion–extensions that the participant was able to perform in 30 s with a 2.5 kg dumbbell. 

The Six-Minute Walk Test is an assessment of endurance, which measures the meters that the participant is able to cover walking in a circuit in six minutes.

The anthropometric measures, the EEPAQ questionnaire and the physical condition variables were evaluated at the beginning of the study and 24 months later. During this period, the participants followed the training plan described below.

### 2.5. Training

All the women who participated in the study received training from a professional who was qualified in physical activity and sports. This training consisted of two weekly sessions lasting one hour, and it was based on elderly physical activity guidelines in all municipalities. All programmes included a warm up based on mobility and cardiorespiratory exercises, followed by endurance, resistance training and balance training in each session. Muscle strengthening included standing and sitting on a chair (similar to a squat exercise but modified accordingly to each person’s capacity), quadriceps and calf muscles exercises to improve walking, and upper body exercises such as biceps curls and triceps and shoulder exercises—all of these with light weights. Core exercises were also included. Finally, a cool down based on low-intensity cardiorespiratory exercises followed by flexibility exercises was performed. The intensity of effort during training sessions was based on the Perceived Exertion Scale (BORG scale). The scale was visible for all (displayed on a poster in the room) and used during the session.

All participants included in the study performed the training program for 24 months. This study only included women who, after two years of training, had a minimum of 80% adherence to the program. 

### 2.6. Genotyping

The genomic DNA was obtained via buccal swabs using a standardized procedure of extraction with phenol/chloroform and subsequent precipitation in alcohol in the presence of salts [30].

The genotyping of the *ACTN3 R577X* polymorphism was performed by amplifying a fragment of 303 pairs of bases (bp) via PCR (Polymerase Chain Reaction) using the following primers: *ACTN3*-F 5′-CTG TTG CCT GTG GTA AGT GGG y *ACTN3*-R 5′-TGG TCA CAG TAT GCA GGA GGG. Thirty-five cycles of PCR amplification were performed with an annealing temperature of 60 °C.

The detection of the *R577X* polymorphism was performed via the enzymatic digestion of the amplicons generated by PCR with D de I (Biolabs) and subsequent visualization via electrophoresis in non-denatured 8% acrylamide gels. 

The detection of the *I/D* polymorphism of *ACE* was performed by amplification with PCR using the following primers: 5′-TGGAGAGCACTCCCATCCTTTCT and 5′-GACGTGGCCATCACATTCGTCAGAT. The PCR amplification was performed by use of 35 cycles and an annealing temperature of 58 °C. The result of the resulting PCR was visualized using agarose gel 1.5% with ethidium bromide.

To ensure the correct genotyping of the *DD* cases and avoid cases in which the *DD* genotype could be confused with *ID*, a new nested PCR was used [31] with the following primers: 5′-TGGGACCACAGGCGCCCGCCACTAC and 5′-TCGCCAGCCCTCCCATGCCCATAA. The PCR conditions were previously described, but with an annealing temperature of 64 °C. 

### 2.7. Statistical Analysis

The data obtained from the present study were analyzed using the SPSS 23.0 (SPSS Inc., Chicago, IL, USA) statistical program. Descriptive analyses were performed considering measures of central tendency, asymmetry measures and measures of shape. A normality test was performed using the Kolmogorov–Smirnov criteria. 

A quantitative study was performed to assess the influence of the genetic variables with the remaining parameters under study. As the variables did not display normal behavior, the statistical analysis was performed via the non-parametric Kruskal–Wallis test and the Mann–Whitney U test. The Wilcoxon ranges test was used for the study of related variables.

A 95% confidence interval was used. Likewise, the Tukey test was used to compare differences between the three genotypes of *ACE* and *ACTN3*. To evaluate the Hardy–Weinberg equilibrium of the sample, the Chi-squared test was used.

## 3. Results

Samples of 296 women were gathered, from which the optimal DNA was obtained for amplification and *ACTN3* genotyping. However, due to technical problems, the determination of the *ACE* genotype could only be performed in 282 samples. The minimum age was 60 and the maximum was 91 with an average age of 73.62 (±5.4).

The distribution of alleles the variants was adjusted to the Hardy–Weinberg equilibrium for both genes (*ACTN3*: χ^2^ = 0.79; *p* = 0.37; *ACE*: χ^2^ = 0.23; *p* = 0.63).

The distribution of both genes was as follows:*ACTN3*: *RR* = 93 (31.4%); *RX* = 139 (47%); *XX* = 64 (21.6%). 
*ACE*: *DD* = 105 (37.2%); *ID* = 131 (46.5%); *II* = 46 (16.3%).

Allelic frequencies for ACE and *ACTN3* variants are described for European population as *ACE*: *D* = 0.642, *I* = 0.358; and *ACTN3*: *R* = 0.566, *X* = 0.434 (1000 genomes project). In this research, the allelic frequencies were very similar (*ACE D*: 0.604; *I*: 0.395 and *ACTN3 R*: 0.549; *X*: 0.451). 

This study evaluated the effect of both genotypes on the anthropometric variables under study and on the EEPAQ questionnaire. No significant differences were observed in any of the analyzed variables attributable to the *ACTN3* genotype or *ACE*, neither at the beginning nor at the end of the training. The time dedicated to sedentary activities increased in all groups after training. *ACTN3* heterozygotes significantly increased weight (*p*: 0.039) and BMI (*p*: 0.008) scores after training. The mean and standard deviation of the anthropometric data and EEPAQ questionnaire were as follows: BMI 28.52 (±3.9) before training (BT) and 28.74 (±4.2) after training (AT); % fat mass 39,1 (±4.8) BT and 39.1 (±4.9) AT; hours walking per day 1.83 (±0.8) BT and 1.87 (±0.9) AT; hours sitting per day 3.97 (±1.3) AT and 4.48 (±1.4).

### 3.1. Results Associated to ACE I/D Polymorphism

Table 1 displays the values of physical activity at baseline and after training and its association with training and genotypes. The Chair Stand Test did not obtain significant results either at the intragroup level (because of training effects) or at the intergroup level (because of differences between the genotypes). 

In the case of the Arm Curl Test, all groups improved their results. In this test, after training, a significant result was obtained for the carriers of the allele *D* in both arms; for the genotype *ACE II*, this was significant in the case of the left arm.

In the endurance test (Six-Minute Walk Test), all genotypes worsened their values after training. 

When grouping the genotypes as *II+ID* vs. *DD* or *II* vs. *DD+ID*, no results were found attributable to the genotype, which was also the case regarding the initial moment (T1) or after training (T2) in any of the tests; all groups experienced improved results in the Arm Curl Test after training, whereas the Six-Minute Walk Test scores significantly worsened.

### 3.2. Results Associated to ACTN3 R577X Polymorphism

The study findings are shown in Table 2, revealing that training had a statistically significant effect on the Chair Stand Test: the carriers of genotype *ACTN3 XX* displayed significantly improved results compared to the other genotypes.

On the intragroup level, statistically significant changes were observed in all the tests. In the Arm Curl Test, after two years of training, the carriers of the allele X improved significantly with one or more copies.

In the Six-Minute Walk Test, all groups worsened, despite the training.

When grouping by genotypes as *RR* vs. *RX + XX* or *XX* vs. *RX + RR*, the *X* allele obtained significant improvements after training in the Arm Curl Test. In endurance, significant changes were obtained and worse in both groups after training.

### 3.3. Combined Effect of ACE and ACTN3

Lastly, the genotypes of both polymorphisms were grouped, distinguishing between “POWER” (*ACE DD + ACTN3 RR/RX*); and “NON-POWER” (*ACE II/ID + ACTN3 XX*) [15,22,32]. The results obtained in the tests, before and after training, are displayed in Table 3. Statistically significant differences were obtained, post training, for the non-power group in the Chair Stand Test; once again, a worsening was observed in both groups in the Six-Minute Walk Test.

## 4. Discussion

This study attempts to evaluate the effect of two genetic polymorphisms, *ACE I/D* and *ACTN3 R577X*, among a large number of female active subjects above the age of 60 after a two-year training program. According to the literature consulted, this is the largest study performed for both polymorphisms among this type of population, with regards to both the number of subjects and the training time.

The studies performed on older people have shown varied results according to multiple variables, for which age, gender and lifestyle are differentiating elements. For this reason, this study was solely based on older women, aged 60 years or older, and with an active lifestyle.

The results obtained attribute a favorable effect to the *ACTN3 XX* genotype in the Chair Stand Test and the Arm Curl Test after a two-year training program; however, a generalized worsening was found for all participants in the Six-Minute Walk Test. In addition, no differences were found in the *I/D* polymorphism of *ACE* among the different groups either before or after training. 

This study has analyzed two polymorphisms that are extensively studied in the sports literature in relation to two very important parameters of physical condition in older people: strength and endurance. We found different responses according to genotype for the tests evaluating strength. However, in the tests that monitored endurance, a generalized worsening was observed, despite the training, with no relation to the genotypes studied.

Both polymorphisms continue to be a source of study in numerous studies in the field of sports, especially in athletes [33,34]. However, in the case of the older population, many external factors can affect a person’s physical condition, constraining different results to those obtained in professional athletes. Overall, it is striking that, despite the training and considering the participants as being active people, we found a worsening in the endurance test (Six-Minute Walk Test); nonetheless, it is important to consider that the total sedentary time increased after the two study years. 

To our knowledge, no previous research has been undertaken on older people studying the relationship between the analyzed genotypes and the Six-Minute Walk Test after training. Giacaglia et al. [23] used the same test over an 18-month training period and assessed the relationship with *I/D* polymorphism *ACE*, without finding changes. A study by Moraes et al. [35] also failed to find significant differences attributable to *ACE* polymorphism, with participants only improving their results after a 12-week training period. Regarding *ACTN3 R577X* polymorphism, the results obtained indicate that, despite the poor results among all groups, the *RR* homozygotes significantly worsened more than the carriers of the *X* allele. 

Related to the muscle strength of participants, in the Chair Stand Test, noteworthy changes were found due to training and the *ACTN3* genotype. Thus, women with the *ACTN3 XX* genotype present improved results compared to the remaining groups after the 24-month training period. In the combined analysis of genes, improved results were found, once again, in the “NON-POWER” group which included this genotype (*ACE II/ID + ACTN3 XX*) compared to the “POWER” group (*ACE DD + ACTN3 RR/RX*). Several studies in which the training program was not performed but where the same strength test was used in older women [15,36,37] failed to find a relationship with the *ACTN3* genotype, which is similar to the results found in the first measurement performed. Kikuchi et al. [38] also performed the Chair Stand Test, without training, and failed to find a difference attributable to the genotype in the group of women analyzed. In contrast to our findings, Pereira et al. [32] observed differences in both genotypes of this same test favorable to the *RR* homozygotes of *ACTN3* and to the *ACE DD* group after a 12-week training period. In this case, the participants performed a shorter training, the mean age was lower, and there were significant differences between genotypes and body composition, all of which may justify the discrepancies between these findings. Due to contradictory results in studies that have reviewed the effects of *ACTN3* after training [39], in this study, we have tried to homogenize the characteristics of the participants in terms of the training received, age, sex and lifestyle. However, the study by Seto et al. [40] in mice could explain why better results have been found in homozygotes *XX* since the absence of α-actinin-3 seems to produce an increase in calcineurin activity, producing a reprogramming of the metabolic phenotype of the fast muscle fibres and causing the muscle to have a greater adaptation response to training.

To evaluate upper limb strength, our study used the Arm Curl Test. We were only able to find one similar study performed in older people and measuring this variable with the same test used in this study [36]; no significant differences were found which were attributable to genotype, although an improvement was found in all groups due to training. Likewise, other research groups who evaluated upper limb strength using other tests have also failed to find a relationship with *ACE* over time [26,41,42].

Concerning the *ACTN3* polymorphism and the Arm Curl Test, improved results were obtained after training for the *XX* homozygotes in the left arm. In the right arm, significant improvements were observed for all the carriers of the *X* allele after training. In line with these findings, Clarkson et al. [43], in a study involving a group of young women, found improved results for the group of *XX* homozygotes after a 12-week training. However, other authors, such as Delmonico et al. [44], obtained differences over time; in this case, without any training.

## 5. Conclusions

The following conclusions may be drawn, based on the analysis of both polymorphisms in this group of the population: first, on a general level, over these two years, despite leading an active lifestyle, the women in our study increased the amount of time dedicated to sedentary activities and have worsened their endurance capacity. Second, the analysis of *I/D* polymorphism of *ACE* does not seem to have a very relevant effect on the tests studied over time. Furthermore, the *ACTN3 R577X* polymorphism could have an important role in capacities related to muscle strength, providing a beneficial effect for carriers of allele *X*. Lastly, the combination of both polymorphisms (“POWER” vs. “NON-POWER”) does not provide additional advantages to those achieved studying the non-combined genotypes.

## Figures and Tables

**Table 1 ijerph-17-01236-t001:** Physical tests at baseline and after training and its association with training and *ACE* genotypes.

Test	Genotype	T1 Mean & SD	T2 Mean & SD	*p*	*p*	*p*
				Training Effect	Genotype Effect T1	Genotype Effect T2
Chair Stand Test	*ACE DD* (*n* = 105)	14.31 ± 3.2	14.15 ± 3.2	0.71	0.409	0.665
	*ACE ID* (*n* = 131)	1495 ± 3.2	14.57 ± 2.8	0.59		
	*ACE II* (*n* = 46)	14.76 ± 3.2	14.51 ± 2.7	0.897		
	*ACE ID + II* (*n* = 177)	14.90 ± 3.2	14.55 ± 2.7	0.687	0.185	0.421
	*ACE DD + ID* (*n* = 136)	14.67 ± 3.2	14.38 ± 3.0	0.876	0.777	0.915
Right Arm Curl Test	*ACE DD*	17.12 ± 3.5	17.76 ± 3.8	0.026 *	0.988	0.72
	*ACE ID*	17.24 ± 3.5	18.08 ± 3.6	0.003 *		
	*ACE II*	17.32 ± 3.5	18.03 ± 3.8	0.208		
	*ACE ID + II*	17.26 ± 3.5	18.06 ± 3.6	0.002 *	0.978	0.42
	*ACE DD + ID*	17.19 ± 3.5	17.93 ± 3.6	<0.001 *	0.88	0.723
Left Arm Curl Test	*ACE DD*	17.59 ± 3.7	18.59 ± 3.9	0.006 *	0.99	0.71
	*ACE ID*	17.67 ± 3.4	18.23 ± 3.5	0.038 *		
	*ACE II*	17.63 ± 3.6	18.78 ± 3.4	0.047 *		
	*ACE ID + II*	17.66 ± 3.4	18.37 ± 3.4	0.005 *	0.926	0.592
	*ACE DD + ID*	17.64 ± 3.5	18.39 ± 3.7	0.001 *	0.892	0.683
Six-Minute Walk Test	*ACE DD*	531.04 ± 76.1	508.17 ± 89.5	0.012 *	0.313	0.346
	*ACE ID*	536.01 ± 64.2	498.03 ± 95.9	<0.001 *		
	*ACE II*	547.60 ± 76.4	510.99 ± 90.3	0.013 *		
	*ACE ID + II*	538.97 ± 67.5	501.37 ± 94.4	<0.001 *	0.522	0.556
	*ACE DD + ID*	533.83 ± 69.5	502.62 ± 93.0	<0.001 *	0.129	0.293

* *p* < 0.05. T1: baseline. T2: after training. SD: standard deviation

**Table 2 ijerph-17-01236-t002:** Physical tests at baseline and after training and its association with training and *ACTN3* genotypes.

Test	Genotype	T1 Mean & SD	T2 Mean & SD	*p*	*p*	*p*
				Training Effect	Genotype Effect T1	Genotype Effect T2
Chair Stand Test	*ACTN3 RR* (*n* = 93)	14.88 ± 3.4	14.41 ± 3.4	0.890	0.853	0.024 *
	*ACTN3* RX (*n* = 139)	14.74 ± 3.0	14.03 ± 2.9	0.061		
	*ACTN3* XX (*n* = 64)	14.46 ± 3.1	15.26 ± 2.1	0.020 *		
	*ACTN3 RX+XX* (*n* = 203)	14.66 ± 3.0	14.44 ± 2.7	0.852	0.873	0.979
	*ACTN3 RR+RX* (*n* = 232)	14.80 ± 3.2	14.19 ± 3.1	0.289	0.574	0.011 *
Right Arm Curl Test	*ACTN3 RR*	16.85 ± 3.5	17.39 ± 3.9	0.051	0.360	0.272
	*ACTN3 RX*	17.51 ± 3.3	18.22 ± 3.6	0.010 *		
	*ACTN3 XX*	17.02 ± 3.7	18.78 ± 4.0	0.005 *		
	*ACTN3 RX + XX*	17.36 ± 3.4	18.41 ± 3.7	<0.001 *	0.309	0.165
	*ACTN3 RR + RX*	17.24 ± 3.4	17.87 ± 3.8	0.001 *	0.561	0.200
Left Arm Curl Test	*ACTN3 RR*	17.18 ± 3.6	17.75 ± 3.6	0.069	0.375	0.070
	*ACTN3 RX*	17.96 ± 3.3	18.52 ± 3.7	0.017 *		
	*ACTN3 XX*	17.72 ± 3.7	19.61 ± 3.9	0.004 *		
	*ACTN3 RX + XX*	17.89 ± 3.4	18.89 ± 3.8	<0.001 *	0.235	0.079
	*ACTN3 RR + RX*	17.65 ± 3.5	18.20 ± 3.6	0.003 *	0.776	0.041 *
Six-Minute Walk Test	*ACTN3 RR*	535.05 ± 67.8	492.32 ± 91.3	<0.001 *	0.747	0.115
	*ACTN3 RX*	535.86 ± 76.6	510.86 ± 82.1	0.004 *		
	*ACTN3 XX*	545.32 ± 63.4	508 ± 110.8	0.003 *		
	*ACTN3 RX + XX*	538.80 ± 72.7	510.05 ± 92.3	<0.001 *	0.640	0.049 *
	*ACTN3 RR + RX*	535.54 ± 73.1	503.23 ± 86.3	<0.001 *	0.466	0.172

* *p* < 0.05. T1: baseline. T2: after training. SD: standard deviation

**Table 3 ijerph-17-01236-t003:** Effect of training and combination of *ACTN3* and *ACE* genotype on physical tests.

Test	Grouped Genotype	T1 Mean & SD	T2 Mean & SD	*p*	*p*	*p*
				Training Effect	Genotype Effect T1	Genotype Effect T2
Chair Stand Test	POWER (*n* = 82)	14.43 ± 3.1	13.77 ± 3.3	0.617	0.461	0.031 *
	NON-POWER (*n* = 38)	14.86 ± 2.9	15.09 ± 1.8	0.505		
Right Arm Curl Test	POWER	17.19 ± 3.2	17.22 ± 3.5	0.192	0.778	0.309
	NON-POWER	17.22 ± 3.3	18.14 ± 3.7	0.154		
Left Arm Curl Test	POWER	17.62 ± 3.6	17.92 ± 3.7	0.115	0.885	0.410
	NON-POWER	17.86 ± 3.5	18.80 ± 3.4	0.124		
Six-Minute Walk Test	POWER	529.05 ± 79.3	504.77 ± 83.5	0.035 *	0.354	0.793
	NON-POWER	546.88 ± 64.0	497.46 ± 116.0	0.018 *		

* *p* < 0.05. T1: baseline. T2: after training.

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
