# Peer review of "Strength and Endurance Training in Older Women in Relation to *ACTN3 R577X* and *ACE I/D* Polymorphisms"

_ijerph, 2020, doi:10.3390/ijerph17041236_

Round 1

Reviewer 1 Report

The article titled "Strength and resistance training in older women in relation to ACTN3 R577X and ACE I/D polymorphisms" evaluates the relationship among different genotypes of the 2 polymorphisms ACTN3 and ACE on strength and resistance training of elderly women.

The large sample and the motor test applied make the research original.

The manuscript, in my opinion, can be accepted for publication after minor revision, in details:

1) In materials and method, please better define the sampling, in particular specify the nationality and the exact provenience of the individuals (the whole Spain?). If some strict criteria were adopted (born in the region for three generations) they should be specified.

2) Has it verified that the sample had a homogeneous motor activity back ground? I mean that in the sample were not present together ex-athlete and sedentary people? I think that this could affect the results.

3) I appreciate the use of nest PCR, but I have just the curiosity to know why you use it only for ACE, has it more ambiguous results?

4) in results: I would be nice to add that genotype and allele frequencies obtain fall in the variation range of European population (for example, 1000 genome data can be used)

5) in results: I think that the calculation of ODDS Ratio would increase the statistic power of the results.

6) in results - Combined effect of ACE and ACTN3: I think that the collocation of heterozygotes in power or not power is quite arbitrary. The exact collocation of heterozygote can be clarified applying the recessive or dominant model of Odds ratio.

7) in discussion: since most of the researches on sport performance reported that ACTN3 null allele homozygotes (577XX) were underrepresented in power athletes, and the ACTN3R allele is associated with enhanced improvements in strength (e.g. Pickering and Kiely 2017; Massidda et al., 2012), maybe the physiological mechanism of action of ACTN3 should be better explain, because results seems contradictory with previous ones.

Author Response

REVIEWER 1

In materials and method, please better define the sampling, in particular specify the nationality and the exact provenience of the individuals (the whole Spain?). If some strict criteria were adopted (born in the region for three generations) they should be specified.

The following text has been included: “The EXERNET multi-center study is a cross-sectional study on physical fitness and body composition evaluation and its relation with healthy lifestyle among non-institutionalized elderly from 6 regions in Spain. The population was selected by means of a multi-step, simple random sampling, taking into account, first, the location that ensured the geographic and cultural diversity of the sample, then 3 different cities of each region (the capital of the region and two other cities; one of 10,000–40,000 habitants and another of 40,000–10,000 habitants) and, finally, by random assignment of the civic and sports centers. All participants were of the same Caucasian (Spanish) descent from three or more generations.”

Has it verified that the sample had a homogeneous motor activity back ground? I mean that in the sample were not present together ex-athlete and sedentary people? I think that this could affect the results.

In order to assess the homogeneity of the sample, anthropometric measurements and the EEPAQ questionnaire were carried out and it was checked that there were no differences between the groups so as not to affect the results.  However, their athletic or sedentary status prior to inclusion in the study was not assessed.

I appreciate the use of nest PCR, but I have just the curiosity to know why you use it only for ACE, has it more ambiguous results?

As it is stated in the manuscript, Shanmugam et al described in 1993 that the amplification of the I allele of ACE is sometimes suppressed in an ID heterozygote so that the latter can be mistyped as DD. To discriminate these two conditions, a nested PCR was recommended on this work, the same protocol that was used in the manuscript sent for publication.

The analysis of ACTN3 by restriction fragment length analysis was conclusive in all conditions, so did not need any additional approach.

in results: I would be nice to add that genotype and allele frequencies obtain fall in the variation range of European population (for example, 1000 genome data can be used)

The following text has been included: “Allelic frequencies for ACE and ACTN3 variants are described for European population as ACE: D=0.642, I=0.358; and ACTN3: R=0.566, X=0.434 (1000 genomes Project). In this research allelic frequencies were very similar (ACE D: 0,604; I: 0,395 and ACTN3 R: 0,549; X: 0,451).”    

in results: I think that the calculation of ODDS Ratio would increase the statistic power of the results.

We welcome the suggestion to increase the statistical power of the results. When we analysed the results, we thought that, in order to calculate the ODDS Ratio we should categorize the variables in an arbitrary way losing statistical power and even introducing a possible bias, that is why it was not included in our study. If you think that we should proceed with the calculation or handle the data in another way, we will leave that up, and we will perform the relevant calculations.

6) in results - Combined effect of ACE and ACTN3: I think that the collocation of heterozygotes in power or not power is quite arbitrary. The exact collocation of heterozygote can be clarified applying the recessive or dominant model of Odds ratio.

The collocation of the heterozygotes has been done following other authors and after reviewing the literature. We want to apologize because there was an error in the naming of the groups that has been corrected and the citation of these studies has been included.

in discussion: since most of the researches on sport performance reported that ACTN3 null allele homozygotes (577XX) were underrepresented in power athletes, and the ACTN3R allele is associated with enhanced improvements in strength (e.g. Pickering and Kiely 2017; Massidda et al., 2012), maybe the physiological mechanism of action of ACTN3 should be better explain, because results seems contradictory with previous ones.

The following text has been included to explain this aspect:

“Due to contradictory results in studies that have reviewed the effects of ACTN3 after training (Pickering and Kiely 2017), in this study we have tried to homogenize the characteristics of the participants in terms of the training received, age, sex and lifestyle. However, the study by Seto et al in mice could explain why better results have been found in homozygotes XX since the absence of α-actinin-3 seems to produce an increase in calcineurin activity producing a reprogramming of the metabolic phenotype of the fast muscle fibres and making the muscle have a greater adaptation response to training.”

Reviewer 2 Report

Abstract/Introduction

The purpose of the study should be rephrased. The reviewer suggests as follows:

The purpose of this study is to analyze the effect of two genetic polymorphisms: ACTN3 R577X, and ACE I/D, on physical condition in a sample of active older women after a two-year training period, Or, The purpose of this study is to evaluate the association of two genetic polymorphisms: ACTN3 R577X, and ACE I/D with strength and resistance values in a sample of active older women, after a two-year training period.

Please keep in mind, that gene names should be italicized – it is to be done throughout the manuscript (e.g. page 1 line 43 ,44; page 2, line 46, 48, 57, 61).

Page 1, line 44 – the phrase “…the I/D polymorphism (rs1799752) of the angiotensin-converting enzyme” should end with “gene”.

Page 2, line 45 – phrase ”…ACTN3 is a structural component protein…” is not proper, the name for protein is α-actinin 3.

Page 2, line 58 – Sentence “More specifically, I alleles have been related with resistance and D alleles have been related…” – should sound “More specifically, I allele has been related with resistance and D allele has been related…

Materials and Methods

2.1. Participants

What was the mean age of participants?

2.5. Training

This part of the manuscript should be described more extensively, because it is a key element of the experiment affecting the obtained results. It is said (page 2, line 77-78)  that subjects participated in a local physical activity program in their  municipality, therefore a few questions arise:

To what extent the training programs implemented in the study were comparable between municipalities? On what basis the intensity of effort during training sessions was calculated? Did participants wear measuring devices (sports testers) analyzing their heart rate to determine the load? What were muscle strengthening exercises?

2.6 Genotyping

What was used to visualize the results of ACTN3 R577X PCR digestion?

Results

There is no complete correlation data for anthropometric as well as EEPAQ questionnaire results with genetic data

Author Response

REVIEWER 2

Abstract/Introduction

The purpose of the study should be rephrased. The reviewer suggests as follows:

The purpose of this study is to analyze the effect of two genetic polymorphisms: ACTN3 R577X, and ACE I/D, on physical condition in a sample of active older women after a two-year training period, Or, The purpose of this study is to evaluate the association of two genetic polymorphisms: ACTN3 R577X, and ACE I/D with strength and resistance values in a sample of active older women, after a two-year training period.

Please keep in mind, that gene names should be italicized – it is to be done throughout the manuscript (e.g. page 1 line 43 ,44; page 2, line 46, 48, 57, 61).

Page 1, line 44 – the phrase “…the I/D polymorphism (rs1799752) of the angiotensin-converting enzyme” should end with “gene”.

Page 2, line 45 – phrase” … ACTN3 is a structural component protein…” is not proper, the name for protein is α-actinin 3.

Page 2, line 58 – Sentence “More specifically, I alleles have been related with resistance and D alleles have been related…” – should sound “More specifically, I allele has been related with resistance and D allele has been related…

We appreciate the suggestion. It has been modified following your comments.

Materials and Methods

2.1. Participants

What was the mean age of participants?

The minimum age was 60 and the maximum was 91 with an average age of 73.62 (±5.4).

This data has been included in the results section.

2.5. Training

This part of the manuscript should be described more extensively, because it is a key element of the experiment affecting the obtained results. It is said (page 2, line 77-78) that subjects participated in a local physical activity program in their municipality, therefore a few questions arise:

To what extent the training programs implemented in the study were comparable between municipalities?

The training programme was based on elderly physical activity guidelines in all municipalities. All programmes included in each session a warm up based on mobility and cardiorespiratory exercises, followed by endurance, resistance training and balance training. Finally, a cool down based on low intensity cardiorespiratory exercises followed by flexibility exercises.

On what basis the intensity of effort during training sessions was calculated?

It was based on the Perceived exertion scale (BORG scale). The scale was visible for all (poster in the room) and used during the session.

Did participants wear measuring devices (sports testers) analyzing their heart rate to determine the load?

No, they did not.

What were muscle strengthening exercises?

They were resistance training exercises including standing and sitting on a chair (similar to a squat exercise but modified accordingly to each person’s capacity), quadriceps and calf muscles exercises to improve walking,  and upper body exercises such as biceps curl, triceps and shoulder, all of these with light weights.  Core exercises were also included.

This part of the manuscript will have the following text:

“All the women who participated in the study received training from a professional who was qualified in physical activity and sports. This training consisted of two weekly sessions lasting one hour, and it was based on elderly physical activity guidelines in all municipalities. All programmes included in each session a warm up based on mobility and cardiorespiratory exercises, followed by endurance, resistance training and balance training. Muscle strengthening included standing and sitting on a chair (similar to a squat exercise but modified accordingly to each person’s capacity), quadriceps and calf muscles exercises to improve walking, and upper body exercises such as biceps curl, triceps and shoulder, all of these with light weights.  Core exercises were also included. Finally, a cool down based on low intensity cardiorespiratory exercises followed by flexibility exercises. The intensity of effort during training sessions was based on the Perceived Exertion Scale (BORG scale). The scale was visible for all (poster in the room) and used during the session.

All participants included in the study performed the training program for 24 months. This study only included women who, after two years of training, had a minimum of 80% adherence to the program.”

2.6 Genotyping

What was used to visualize the results of ACTN3 R577X PCR digestion?

PCR fragments were separated in 1.5% agarose gels (I/D polymorphism of ACE) or 8% non-denaturing acrylamide gels (R577X polymorphism of ACTN3). In all cases, DNA was visualized under ethidium bromide staining, added either during gel preparation (agarose gels) or post-run (acrylamide gels) 

Results

There is no complete correlation data for anthropometric as well as EEPAQ questionnaire results with genetic data.

Data relating to anthropometric measurements and the EEPAQ questionnaire did not obtain significant results in relation to genotype and therefore have not been included in tables.

The following text has been included “Mean and standard deviation about anthropometric data and EEPAQ questionnaire were: BMI 28,52 (±3,9) before training (BT) and 28,74 (±4,2) after training (AT); % fat mass 39,1 (±4,8) BT and 39,1 (±4,9) AT; hours walking per day 1,83 (±0,8) BT and 1,87 (±0,9) AT; hours sitting per day 3,97 (±1,3) AT and 4,48 (±1,4).”

Here you can see the data obtained in general and by genotype. If you want us to include any of the tables, we will leave that up to you.

T1 Mean & SD

T2 Mean & SD

p-

Training effect        

p-

Genotype effect T1

p-

Genotype effect T2

BMI (kg/m2)

ACE DD (n= 105)

28,70 ± 4,1

28,90 ± 4,5

0,088

0,607

0,889

ACE ID (n=131)

28,55 ± 3,7

28,73 ± 3,8

0,104

ACE II (n=46)

28,02 ± 3,8

28,46 ± 4,2

0,279

ACTIVITY (hours walking per day) 

ACE DD

1,78 ± 0,9

1,87 ± 1,0

0,291

0,195

0,493

ACE ID

1,90 ± 0,8

1,94 ± 1,0

0,644

ACE II

1,84 ± 0,7

1,76 ± 1,0

0,647

SEDENTARISM (hours sitting per day)

ACE DD

3,94 ± 1,2

4,48 ± 1,5

0,001*

0,741

0,875

ACE ID

3,99 ± 1,4

4,46 ± 1,3

0,001*

ACE II

4,13 ± 1,5

4,54 ± 1,3

0,084

% FAT MASS

ACE DD

39,06 ± 5,0

39,12 ± 5,5

0,953

0,506

0,535

ACE ID

39,33 ± 5,0

39,51 ± 4,8

0,088

ACE II

38,64 ± 4,3

38,59 ± 4,6

0,697

T1 Mean & SD

T2 Mean & SD

p-

Training effect        

p-

Genotype effect T1

p-

Genotype effect T2

BMI (kg/m2)

ACTN3 RR (n= 93)

28,86 ± 4,0

28,90 ± 4,2

0,988

0,543

0,783

ACTN3 RX (n=139)

28,16 ± 3,7

28,52 ± 4,1

0,008*

ACTN3 XX (n=64)

28,84 ± 4,2

28,95 ± 4,4

0,197

ACTIVITY (hours walking per day)

ACTN3 RR

1,79 ± 0,7

1,77 ± 0,9

0,748

0,832

0,487

ACTN3 RX 

1,85 ± 0,8

1,89 ± 0,9

0,58

ACTN3 XX 

1,86 ± 1,0

1,97 ± 1,2

0,248

SEDENTARISM (hours sitting per day)

ACTN3 RR

4,00 ± 1,4

4,28 ± 1,4

0,067

0,108

0,081

ACTN3 RX 

4,08 ± 1,3

4,67 ± 1,3

<0,001*

ACTN3 XX 

3,68 ± 1,3

4,35 ± 1,4

0,002*

% FAT MASS

ACTN3 RR

39,32 ± 4,7

39,14 ± 4,9

0,757

0,891

0,946

ACTN3 RX 

38,96 ± 5,0

39,20 ± 5,1

0,443

ACTN3 XX 

39,17 ± 4,8

39,03 ± 4,9

0,69

Round 2

Reviewer 2 Report

It is recommended that gene and allele symbols are italicized throughout the manuscript, please to do so.